# Cis-regulatory hubs: a new 3D model of complex disease genetics with an application to schizophrenia

Loïc Mangnier[1,2,3,4] , Charles Joly-Beauparlant[4,5], Arnaud Droit[3,4,5] , Steve Bilodeau[3,6,7] , Alexandre Bureau[1,2,3]

**The 3D conformation of the chromatin creates complex networks of noncoding regulatory regions (distal elements) and promoters impacting gene regulation. Despite the importance of the role of noncoding regions in complex diseases, little is known about their interplay within regulatory hubs and implication in multigenic diseases such as schizophrenia. Here we show that cis-regulatory hubs (CRHs) in neurons highlight functional interactions between distal elements and promoters, providing a model to explain epigenetic mechanisms involved in complex diseases. CRHs represent a new 3D model, where distal elements interact to create a complex network of active genes. In a disease context, CRHs highlighted strong enrichments in schizophrenia-associated genes, schizophrenia-associated SNPs, and schizophrenia heritability compared with equivalent structures. Finally, CRHs exhibit larger proportions of genes differentially expressed in schizophrenia compared with promoter-distal element pairs or TADs. CRHs thus capture causal regulatory processes improving the understanding of complex disease etiology such as schizophrenia. These multiple lines of genetic and statistical evidence support CRHs as 3D models to study dysregulation of gene expression in complex diseases more generally.**

## Introduction

The etiology of complex diseases involves a broad range of causal factors, both genetic and environmental, leading to gene expression changes (Vliet et al, 2007; Do et al, 2017). Models currently used in the etiology of complex diseases suggest that most risk variants are located within noncoding regions explaining a large portion of the heritability (Maurano et al, 2012). Indeed, most risk variants are enriched in distal noncoding regions, disturbing the tissue-specific transcriptional program, and therefore playing a key role in disease etiology (Zhang & Lupski, 2015). The difficulty to assign distal

regulatory elements to genes hampered the ability to discover the underlying molecular mechanisms. Consistent with a role of noncoding regions in complex phenotypes, there is also strong evidence on the involvement of 3D chromatin conformation in gene regulation. The 3D genome organization, captured by chromosome conformation assays (van Berkum et al, 2010), revealed the physical proximity between regulatory elements and putative target genes. In addition to chromatin loops connecting promoters to distal noncoding regions (Gorkin et al, 2014; Bouwman & de Laat, 2015; Dekker & Mirny, 2016), the genome is parsed into larger domains including topologically associating domains (TADs) (Dixon et al, 2012) and A/B compartments (Lieberman-Aiden et al, 2009). Interestingly, DNA sequence variations influencing the 3D genome organization are associated with complex disease risks (Gorkin et al, 2019). For example, structural variants disrupting TADs, which are enriched in enhancer-promoter interactions, lead to fused-TADs promoting ectopic promoter-enhancer connections and disruption of the normal transcriptional program (Fudenberg & Pollard, 2019; Melo et al, 2020). However, precisely identifying which genes are affected by a risk variant remains a challenge.

The combination of chromatin interactions and microscopy-based techniques established that groups of genes share the same physical environment (Gizzi et al, 2019). In fact, promoters interact with enhancers inside complex organizations, forming regulatory hub structures (Oudelaar et al, 2019; Campigli Di Giammartino et al, 2020). These hubs exhibit distinct organization from known 3D features, encompassing in most cases few promoters, strongly involved in biological processes (Espinola et al, 2021). In fact, highly interconnected enhancers converge to genes with crucial phenotypic implications, with dynamic enhancer crosstalk at the genome-wide level, occurring more frequently during differentiation (Madsen et al, 2020). Furthermore, super interacting promoters are enriched in lineage-specific genes (Song et al, 2020), known to play a crucial role in diseases, whereas multiple enhancers connected to a promoter lead to phenotypic robustness in environmental or genetic perturbations (Tsai et al, 2019). At the molecular level, enhancers increase the gene activity

[1]Centre de Recherche CERVO, Quebec City, Canada    [2]Département de Médecine Sociale et Préventive, Université Laval, Quebec City, Canada    [3]Centre de Recherche en données Massives de l'Université Laval, Quebec City, Canada    [4]Centre de Recherche du Centre Hospitalier Universitaire de Québec - Université Laval, Quebec City, Canada    [5]Département de Médecine Moléculaire, Université Laval, Quebec City, Canada    [6]Centre de recherche du Centre Hospitalier Universitaire de Québec – Université Laval, Axe Oncologie, Quebec City, Canada    [7]Département de Biologie Moléculaire, Biochimie Médicale et Pathologie, Faculté de Médecine, Université Laval, Quebec City, Canada

Correspondence: steve.bilodeau@crchudequebec.ulaval.ca; alexandre.bureau@fmed.ulaval.ca

through modulation of transcriptional bursting (Fukaya et al, 2016) or indirectly influencing transcription activation (Benabdallah et al, 2019). Interestingly, the organization of genes and noncoding regulatory regions may be pre-established, present in different cells, highly dynamic during differentiation (Rubin et al, 2017; Espinola et al, 2021). However, whether and how promoters and enhancers interacting in hubs are involved in the etiology of complex diseases are still open questions.

Schizophrenia is a complex chronic brain disorder associated with perturbations in the transcriptional programs of neurons (Ruzicka et al, 2020 Preprint). Indeed, schizophrenia is characterized by long-standing delusions and hallucinations strongly reducing life-expectancy (Sullivan et al, 2012). Recent findings suggest that schizophrenia is explained by a polygenic architecture (Smeland et al, 2020), where most of the risk variants are located within non-coding regions. Schizophrenia-risk loci are enriched in active enhancers or promoters in neurons from the adult human frontal lobe (Roussos et al, 2014; Fullard et al, 2018; Girdhar et al, 2018; Hauberg et al, 2020). Also, multiple studies have demonstrated the involvement of 3D organization in the disorder. For example, chromatin loops are enriched in expression quantitative trait locis or schizophrenia-risk variants impacting the proximal gene regulation (Rajarajan et al, 2018). In addition, ultra-rare structural variants in TAD borders lead to gene dysregulations increasing the risk of schizophrenia (Halvorsen et al, 2020). However, the implication of regulatory hubs in the schizophrenia etiology remains to be addressed.

In the present study, we are defining cis-regulatory hubs (CRHs) as 3D structures linking one or more gene promoters to networks of distal elements which capture complex patterns of gene regulation. In neurons, CRHs are strongly enriched in schizophrenia-associated genes, SNPs, and heritability compared with equivalent structures.

## Results

### Promoter and distal element interactions create CRHs in neurons

To understand regulatory processes involved in complex pheno-types, we built CRHs as bipartite graphs, a natural structure for the 3D contacts between two classes of elements: the promoter of genes and their distal elements. To evaluate their role in schizophrenia etiology, we defined CRHs using chromatin contacts provided by Hi-C data with and without additional epigenetic features defining classes of distal regulatory elements (See Supplemental Data 1). Because open chromatin regions in the prefrontal cortex of schizophrenia individuals have shown to be enriched in risk variants (Bryois et al, 2018) and that H3K27ac regions are strongly associated with schizophrenia-risk variants (Girdhar et al, 2018), we focused our attention on the activity-by-contact (ABC) model (Fulco et al, 2019) of enhancer–promoter interactions. The approach integrates the frequency of physical contacts between distal elements and promoters (500 bp from an annotated TSS) with the activity defined by the DNAse accessibility and the occupancy of H3K27ac (Fig 1, see the Materials and Methods section). The ABC model is a good predictor of differential gene expression (Fulco et

al, 2019) and a useful tool to link noncoding variants to their target genes (Nasser et al, 2021).

Using available datasets in neurons derived from induced pluripotent stem cells (iPSCs) (Rajarajan et al, 2018), a relevant cell type to study schizophrenia (Sey et al, 2020), we identified 62,658 functional pairs of distal elements and promoters where the ABC score exceeded a threshold of 0.012. The value of the threshold was chosen so that we had, on average, 4.51 distal elements per genes, as recommended by Fulco et al (2019). CRHs were built from these connections between promoters and distal elements (Fig 1A–C). We identified 1,633 CRHs, ranging between 2 and 506 nodes (median of six elements). Postmortem brains are an alternative source of neurons to study 3D contacts. In three samples from postmortem brains: dopaminergic neurons (mentioned as Dopa_1 and Dopa_2 in the Supplemental Data 1) and general neuron population (mentioned as Neu), respectively (Espeso-Gil et al, 2020, See Supplemental Data 1), we observed a strong overlap of the pairs of promoters and distal elements detected by the ABC approach with those found in iPSC-derived neurons. Indeed, we found that most distal elements were shared between iPSC-derived neurons and postmortem brains (Fig S1A). Also, more than 75% of pairs were either strictly found in iPSC-derived neurons (e.g., identical pair) or in an indirect connection within the same CRH (Fig S1B), supporting the reproducibility of the proposed method.

To start investigating the complexity of CRHs in iPSC-derived neurons, we surveyed genes associated with schizophrenia (Schizophrenia Working Group of the Psychiatric Genomics Consortium, 2014). For example, genes involved in glutamatergic transmission or synaptic plasticity pathways (GRIA1, GRIN2A, and GRM3) exhibited strong differences regarding CRH complexity (Figs 2A and S2–S4). Indeed, GRIN2A and GRM3 were found in relatively simple CRHs of two (one promoter and one enhancer) and three (one promoter and two enhancers) nodes, respectively, whereas GRIA1 was found in a complex network with 24 genes and 72 regulatory regions. Among the 1,633 CRHs, 15% were pairs of two nodes and therefore constituted monogamous relationships, whereas 85% had 3 elements or more (Fig 2B). We observed comparable results in postmortem brains (Fig S5A and B). Moreover, in iPSC-derived neurons and in post-mortem brain tissues, CRHs contained, on average, a significantly higher number of distal elements than promoters, up to twofold more (median of five distal elements against two promoters, two-sided Wilcoxon signed-rank test, $P$-value $\leq 2 \times 10^{-16}$) (Figs 2C and S6). Accordingly, promoters were more connected than distal elements as the 80% least connected promoters had at least twice as many connections as the corresponding 80% distal elements (Fig 2D). This result was confirmed in postmortem brain (Fig S7A–C). As expected, the proportion of distal elements was positively correlated with the connections between promoters and distal elements (or complexity) within CRHs (Spearman $\tau = 0.37$, $P$-value $\leq 2 \times 10^{-16}$), revealing that complex CRHs are significantly associated with a higher proportion of distal elements. The above results suggest that distal regulatory elements and gene promoter regions are organized into complex regulatory structures in neurons.

The connectivity between promoters and enhancers is strongly associated with the emergence of tissue-specific phenotypes as

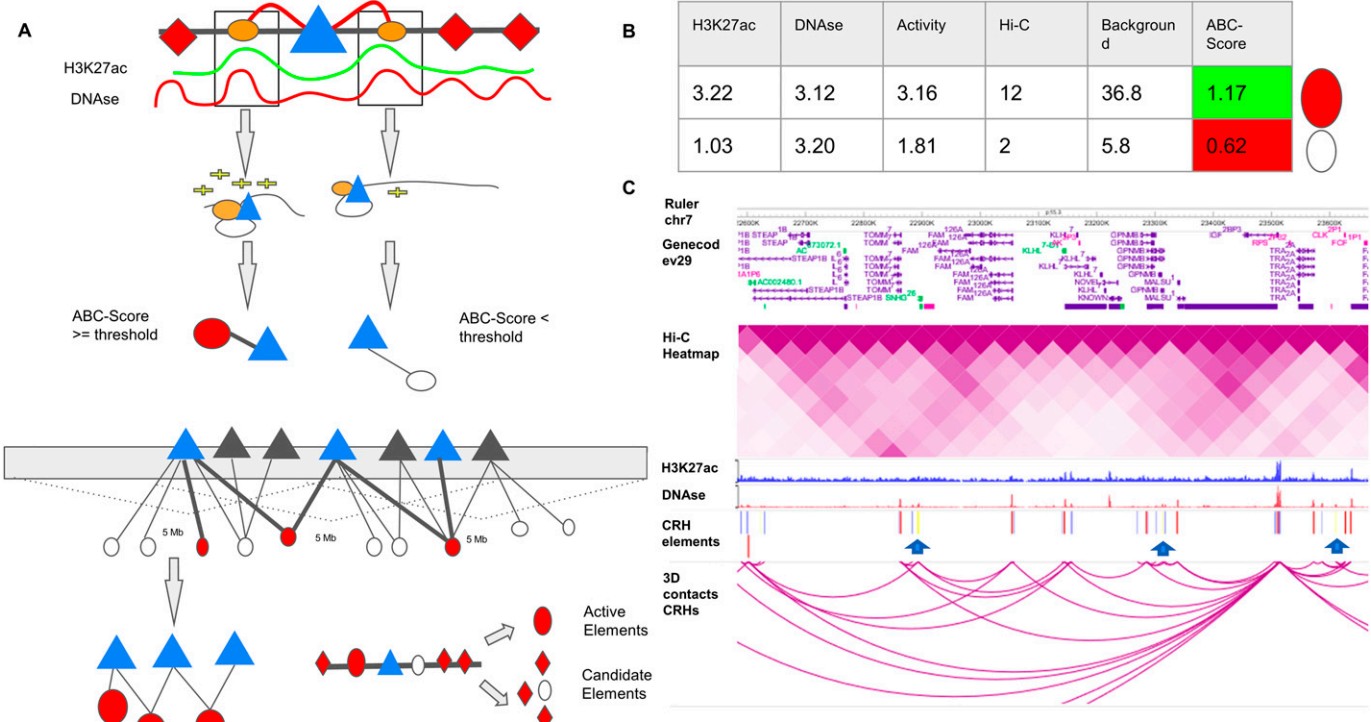

**Figure 1.  Cis-regulatory hubs (CRHs) are built from activity-by-contact (ABC)-Score methodology.**
**(A)** Diagram showing ABC-Score methodology to build functional pairs of promoter (triangles) and distal element (circles). H3K27ac and DNAse signals are shown on an arbitrary scale. Among all distal elements (orange circles), active elements (red circles) are discriminated from candidate elements (white circles) based on the value for the ABC-Score. Candidate elements are DNAse accessible regions without H3K27ac signal and non-overlapping active elements. **(B)** Example for the computation of the ABC-Score. Activity is calculated with geometric mean of H3K27ac and DNAse signals. Finally, the ABC-Score is the product of Activity by Hi-C signal divided by the background activity, within a 5-Mb window. Here we used a threshold of 1.12 to determine functional connection. All values shown in the table are arbitrary.
**(C)** Representation of the physical contacts of a CRH subset on chromosome 7 from the WashU Epigenome Browser (Zhou et al, 2011). Hi-C data in induced pluripotent stem cell–derived neurons are represented. In the CRH element track, we represented distal elements belonging to the CRH (red), promoters (blue) and, elements encompassing noncoding SNPs (yellow bars and blue arrows).

they control the transcriptional program (Tsai et al, 2019). Recent studies have shown that highly connected enhancers converge to genes with strong phenotypic impacts (Madsen et al, 2020), whereas promoters enriched in connections are more tissue-specific (Song et al, 2020). Because we expected that connections of genes or distal elements may play a role in disease emergence, we investigated in more detail the organization of genes and distal elements in CRHs of three nodes or more. Thus, we defined two metrics aiming to characterize genes and distal elements involved in these complex relationships (Fig 2A): (1) the proportion of instances where one distal element connects one promoter with at least one other distal element or the reverse: one promoter connects one distal element with at least one other promoter (i.e., 1-1-N with N > 0) and (2) the proportion of polygamous elements (i.e., which are not in a monogamous pair or 1-1-N, forming complex shared interactions by promoters or distal elements). Interestingly, most distal elements (63%) were connected to a single promoter, whereas 90% of promoters showed interactions with multiple distal elements. We also observed that 1% of promoters are within 1-1-N relationships versus 5% of distal elements (Fig 2E). This result suggests that distal elements share a gene more frequently than genes share a distal element, in accordance with previous findings

in model organisms (Espinola et al, 2021). Also, comparing CRHs built using the ABC approach with other CRH definitions, we found that promoters were also more connected than distal elements (Fig S8A–C). This result was confirmed across postmortem brain tissues (Fig S9). However, promoters showed fewer connections in our other CRH definitions than the ABC approach. Therefore, the proposed definition of CRHs aligns with previous models suggesting that distal elements interact more specifically (Madsen et al, 2020), whereas promoters are more frequent inside complex relationships.

Next, we wanted to determine the relationship between CRHs and known 3D structures. We focused our analysis on A/B compartments (Lieberman-Aiden et al, 2009), TADs (Dixon et al, 2012), and frequently interacting regions (FIREs) (Schmitt et al, 2016), respectively, segmenting the genome into open and close chromatin, domains of frequent interactions between distal elements and genes, and hotspots for chromatin contacts. In iPSC-derived neurons, the majority of CRHs (76%) shared compartments of the same type, with 46% and 29% for active and inactive compartments, respectively (Fig 2F), whereas only a minor portion (8%) of CRHs overlapped several compartments of different types or were in genomic regions not assigned to a compartment (17%). This result

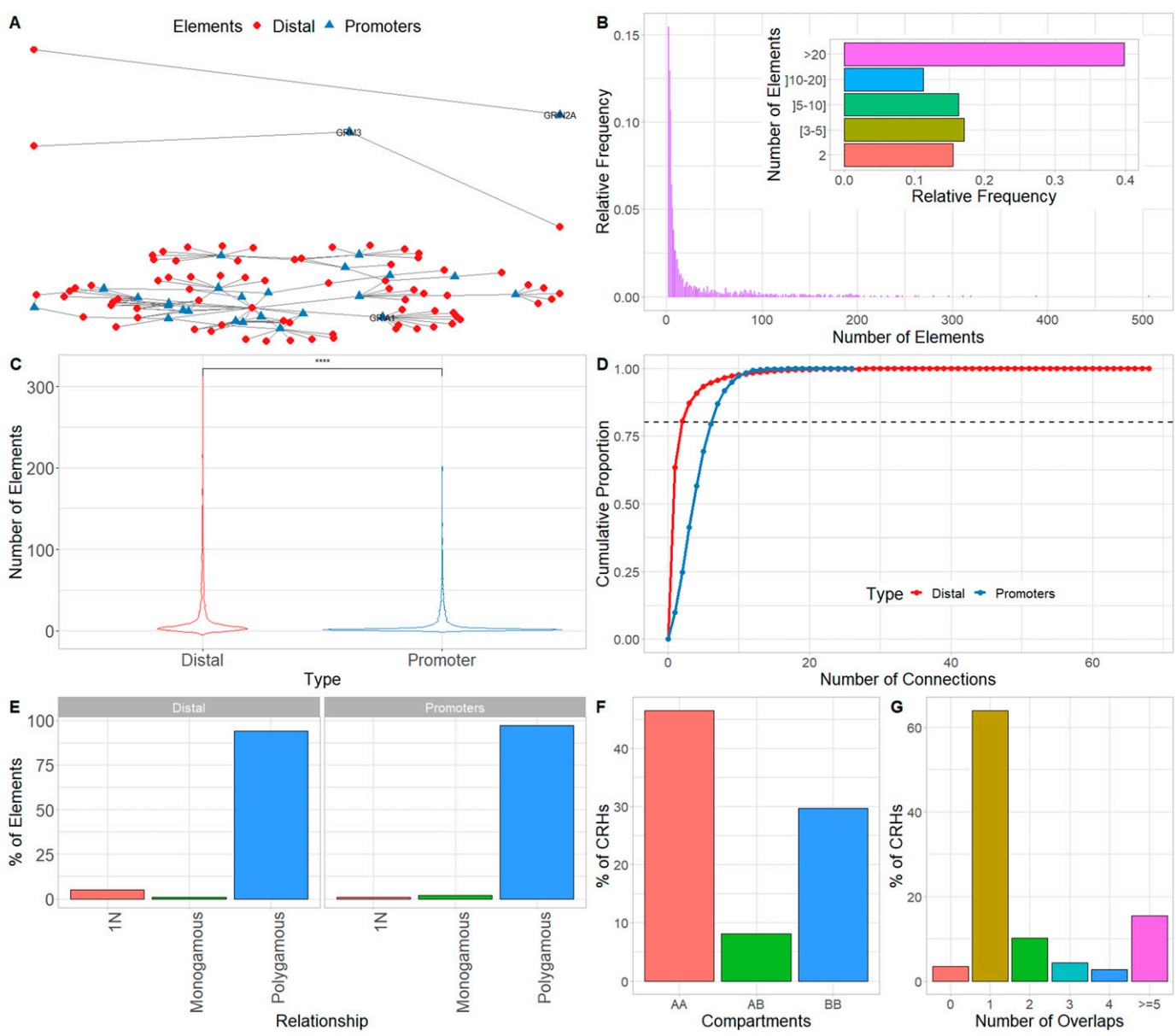

**Figure 2. Cis-regulatory hub (CRH) are 3D-based networks mainly constituted by distal elements and more local than high order 3D features.**
**(A)** CRHs connecting promoters (blue) to distal elements (red) for *GRIN2A* (top), *GRM3* (middle), and *GRIA1* (bottom) genes. The genes are within monogamous, 1-1-N, and polygamous relationships, respectively. Distal elements are represented by red circles, whereas promoters by blue triangles. **(B)** Distribution of the number of elements (promoters and distal elements) within CRHs. The subpanel shows the number of CRH elements by aggregated categories. **(C)** Distribution of the number of promoters (blue) and distal elements (red) per CRH. We used two-sided Wilcoxon signed-rank test to compare the number of elements. **(D)** Cumulative distribution function of the number of connections for promoters (blue) and distal elements (red). The dotted line shows the 80th percentile of the number of connections. **(E)** Distribution of the kind of relationship for distal elements (left) and promoters (right). **(F)** Distribution of overlap of CRHs with each compartment type (AA, Active–Active; AB, Active–Inactive; BB, Inactive–Inactive). When CRHs overlap several compartments, we restrict our attention to the farthest elements. The CRHs in genomic regions not assigned to compartments (17%) were omitted from the distribution. **(G)** Distribution of the number of topologically associating domains overlapped by each CRH, when topologically associating domains are detected with the directionality index. Data information: In (C) **** represents *P*-value ≤ 0.0001.

was confirmed in postmortem brains where 47% and 53% of CRHs overlapped active compartments for general neuronal populations and dopaminergic neuronal nuclei (Fig S10). Because it has been shown that A compartments correlated strongly with the presence of genes, accessible chromatin, activating, and repressive histone marks (Lieberman-Aiden et al, 2009), we argue that CRHs are consistent with the open chromatin characteristic associated with

functional elements. Moreover, most of CRHs (64%) overlapped a single TAD (Fig 2G). Interestingly, 26% of TADs included two or more CRHs. These observations were confirmed in postmortem brain tissues (Fig S11) and by testing multiple TAD detection algorithms (Fig S12A and B). Last, CRHs were enriched in FIREs compared with candidate CRHs (tissue-specific regions non integrating 3D contacts, see the Materials and Methods section) (two-sided Fisher's

exact test, odds ratio = 1.41, $P$-value ≤ $2.2 × 10^{-16}$), although only a minor portion of distal elements or promoters overlapped with FIREs (11% and 13%, respectively). The presence of CRHs within compartments and TADs in addition to the enrichment in FIREs was confirmed using the different CRH definitions (Fig S13A and B). Collectively, our results support that CRHs are networks of interacting regulatory regions and genes at a finer scale than previously defined chromosome structures. Given the similarity between CRHs in neurons from iPSC and from postmortem brain tissue, from now on results are restricted to neurons from iPSC.

## CRHs are defined by active chromatin and the presence of schizophrenia-relevant genes

Genes and regulatory elements sharing the same nuclear environment often show coherent transcriptional states and related molecular functions (Campigli et al, 2020). To further characterize the transcriptional activity of CRHs and their involvement in schizophrenia, we overlaid the chromatin states defined by the Roadmap Epigenomics Consortium (Roadmap Epigenomics Consortium et al, 2015). The 18-states model in neurons was subdivided as follows into three broad categories: (1) Active (1_TssA, 2_TssFlnk, 3_TssFlnkU, 4_TssFlnkD, 5_Tx, 7_EnhG1, 8_EnhG2, 9_EnhA1, 10_EnhA2, and 12_ZNF/Rpts), (2) Weakly Active (6_TxWk, 11_EnhWk, 14_TssBiv, and 15_EnhBiv), and (3) Inactive/Repressor (13_Het, 16_ReprPC, 17_ReprPCWk, and 18_Quies). At the broad category level, we found that most elements (promoters and distal elements) included in CRHs (58%) overlapped Weakly Active regions against 49% for Inactive or Repressor and 53% for Active regions, respectively (Fig 3A). At the individual state level, we observed that 39% of the distal elements included the Quiescent state (Fig 3B) but that CRHs were enriched 2.35-fold (two-sided Fisher's exact test, $P$-value ≤ $2 × 10^{-16}$) in active states and depleted in inactive states (two-sided Fisher's exact test, odds ratio = 0.49, $P$-value ≤ $2 × 10^{-16}$) compared with candidate CRHs. To confirm the enrichment of CRHs in functional elements, we used ENCODE candidate elements in neurons (The ENCODE Project Consortium et al, 2020). ENCODE candidate elements are regions exhibiting significant signals in H3K4me3, H3K27ac, DNAse, or CCCTC-binding factor (CTCF). CRHs were strongly associated with H3K4me3 (two-sided Fisher's exact test, odds ratio = 1.81, $P$-value ≤ $2 × 10^{-16}$), DNAse (two-sided Fisher's exact test, odds ratio = 1.66, $P$-value ≤ $2 × 10^{-16}$), and H3K27ac (two-sided Fisher's exact test, odds ratio = 1.44, $P$-value ≤ $2 × 10^{-16}$), but not with CTCF. Our results were supported by other CRH definitions (Fig S14A and B). Then, to extract the global pattern of chromatin states within a CRH, we kept chromatin states representing up to 80% of the total chromatin state signal and observed a striking difference across CRHs. Indeed, 35% of CRHs exhibited a unique combination of chromatin states (e.g., a set of states found only once in CRHs) (Table S1). Also, CRHs characterized by active states were more complex than those strongly defined by quiescent states (18_Quies) (Fig 3C). Considering the above findings, CRHs are enriched in active distal elements and exhibit a variety of chromatin state combinations, suggesting they are important for the control of the transcriptional program of neurons.

As CRHs are enriched in active elements in neurons, we postulated that they would be enriched in schizophrenia-relevant genes. First, we identified 8,075 genes associated with schizophrenia (False Discovery Rate ≤ 0.05) using H-Magma (Sey et al, 2020), a statistical approach using 3D noncoding regions with genetic data from genome-wide association study for schizophrenia (The Schizophrenia Working Group of the Psychiatric Genomics Consortium et al, 2020 *Preprint*). We found that 35% of genes significantly associated with schizophrenia are within CRHs compared with 23% for all other genes (1.82-fold enrichment, two-sided Fisher's exact test, $P$-value ≤ $2.2 × 10^{-16}$). Moreover, 42% (687/1,633) of CRHs include at least one schizophrenia-associated gene with 23% (376/1,633) harboring several schizophrenia-related genes (mean = 1.77, max = 69) (Fig 3D). Finally, we found that CRHs were enriched in Gene Ontology (GO) biological processes associated with schizophrenia (Fig 3E). Taken together, these results suggest that CRHs are associated with the pathoetiology of schizophrenia, constituting an interesting model for understanding gene regulation and the emergence of complex phenotypes.

## CRHs containing schizophrenia-associated genes are small and highly expressed

To further characterize CRHs including schizophrenia-associated genes, we examined their characteristics regarding complexity and gene expression levels. Interestingly, CRHs encompassing schizophrenia-associated genes showed larger distances between elements than CRHs not harboring schizophrenia-associated genes (Fig 4A). Also, the number of connections with distal elements was slightly lower for schizophrenia-associated genes than non-associated ones (mean associated genes = 4.39, mean non-associated genes = 4.55, two-tailed $t$ test $P$-value = 0.005). In addition, schizophrenia-associated genes included within CRHs showed higher expression levels than non-associated genes (median associated = 6.18, median non-associated = 5.87, two-sided Wilcoxon rank-sum test $P$-value ≤ $2.2 × 10^{-16}$) (Fig 4B) and were enriched in active distal elements (two-sided Fisher's exact test, odds ratio = 1.79, $P$-value ≤ $2.2 × 10^{-16}$). Moreover, schizophrenia-associated genes were more often monogamous genes compared with non-associated ones, showing a 1.34-fold enrichment (two-sided Fisher's exact test, $P$-value = 0.04). Indeed, 26% of monogamous genes are schizophrenia-associated genes against 20% for non-monogamous ones (Fig 4C). The number of distal elements in CRHs harboring schizophrenia-associated genes was correlated negatively with the proportion of associated genes (Spearman $\tau$ = −0.47, $P$-value ≤ $2.2 × 10^{-16}$). These results suggest that schizophrenia-associated genes are, in most cases, within small hubs, less connected to distal elements, but expressed at higher levels than non-associated genes.

## Multivariate analysis of CRH features with respect to schizophrenia-associated genes

To examine the mutually adjusted influence of the factors examined in the previous sections on schizophrenia-associated genes, we fitted a logistic regression of the status of genes (associated versus non-associated with schizophrenia) on RNA level, the number of connections to distal elements, the

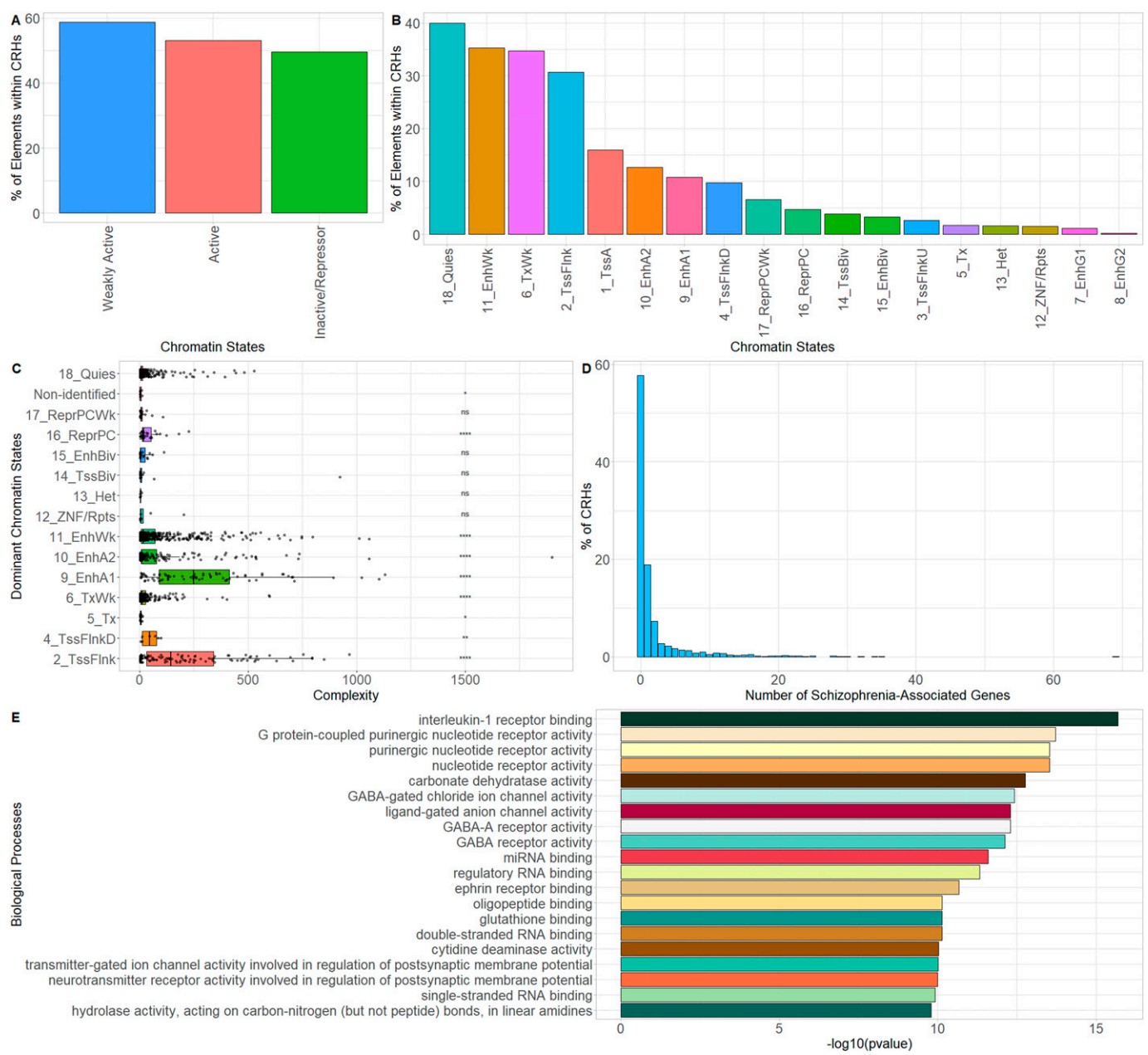

**Figure 3. Cis-regulatory hubs (CRHs) are enriched in transcriptionally active elements and genes associated with schizophrenia.**
**(A)** Proportion of elements (promoters and distal elements) included within CRHs overlapping chromatin states grouped by activity. **(B)** Proportion of elements (promoters and distal elements) included within CRHs overlapping individual chromatin states. **(C)** Boxplot representing complexity by most present chromatin state within each CRH. Two-sided Wilcoxon rank-sum test was used to compare complexity for each chromatin state with 18_Quies state. **(D)** Distribution of the number of schizophrenia-associated genes per CRH. **(E)** GO enrichment for all genes found within CRHs. The top 20 biological processes are represented. Data information: In (D), ns, nonsignificant, * represents $P$-value ≤ 0.05, ** represents $P$-value ≤ 0.01, *** represents $P$-value ≤ 0.001, whereas **** represents $P$-value ≤ 0.0001.

90th percentile of the proportion of active distal elements per gene, and the information regarding monogamy. As expected, the gene status regarding its association with schizophrenia was positively associated with RNA level, the 90th percentile of the proportion of active distal elements, and the monogamy status, whereas it was negatively associated with the number of connections, confirming our results found with univariate analyses (Fig 4D). Collectively, our results suggest that schizophrenia-associated

genes are within small hubs characterized by fewer connections to distal elements and higher transcriptional activity.

## CRHs are enriched in schizophrenia-associated SNPs and heritability

Current models suggest that distal regulatory regions explain a great proportion of the schizophrenia etiology (Roussos et al, 2014).

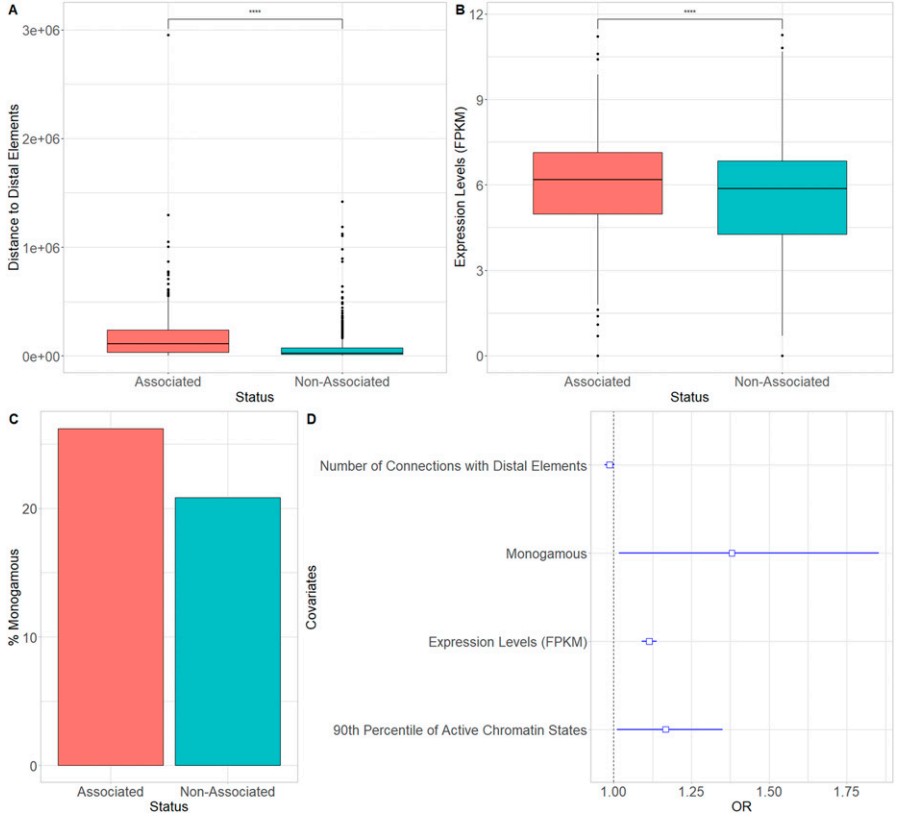

**Figure 4. Features of schizophrenia-associated genes.**
**(A)** Boxplot of mean distance between elements for cis-regulatory hubs encompassing schizophrenia-associated genes and cis-regulatory hubs not harboring ones. Differences were assessed using two-sided Wilcoxon rank-sum test. **(B)** Boxplot of RNA levels for schizophrenia-associated genes and non-associated ones. Differences were assessed using two-sided Wilcoxon rank-sum test. **(C)** Percentage of monogamous genes which are associated with schizophrenia or non-associated. **(D)** Odds ratios (OR) and their 95% confidence interval for a logistic regression of the association status of genes with schizophrenia (yes/no). The dotted line represents the null value. Data information: In (A) and (B) **** represents $P$-values ≤2 × 10$^{-16}$.

In fact, a wide range of genetic variants affecting the gene expression program are involved in the disorder (Huo et al, 2019). Because we demonstrated the enrichment in schizophrenia-relevant genes within CRHs, we next assessed the presence of schizophrenia-associated SNPs. We collected 99,194 SNPs (after clumping, see the Materials and Methods section) from genome-wide association studies (The Schizophrenia Working Group of the Psychiatric Genomics Consortium et al, 2020 *Preprint*). We mapped them to their corresponding CRH and quantified their enrichments at various association $P$-value thresholds using the two-sided Fisher's exact test. For instance, there were 2,058 SNPs with a $P$-value ≤ 1 × 10$^{-4}$. At this significance level, we observed enrichments (odds ratio = 1.29, $P$-value = 0.04) in CRHs compared with the candidate CRHs (Figs 5A and S15A). Then, we used the same methodology as Nasser et al (2021) to define enrichment in common SNPs overlapping a given functional annotation (proportion of significant SNPs for schizophrenia/proportion of all common SNPs). Consistent with our previous finding, we observed higher fold enrichments (enrichment for elements of interest/enrichment for candidates) for CRHs than for distal elements, becoming stronger with the significance level (Fig 5B). This enrichment was stronger with alternative definitions of CRHs (Fig S15B). Therefore, our results suggest that CRHs are enriched in SNPs for schizophrenia.

After demonstrating the relevance of CRHs with schizophrenia-associated SNPs, we wondered whether they explained schizophrenia heritability. To this end, we leveraged linkage disequilibrium score regression (LDSC; Finucane et al, 2015) which provides the portion of disease heritability explained by a functional annotation. First, comparing CRHs to equivalent non-tissue–specific noncoding regions, we ensured to maximize the explained heritability by using tissue-specific elements and integrating 3D contacts by conditioning on enhancers, promoters, H3K27ac, and DNAse peaks from the LDSC baseline model. In addition, we compared CRHs with equivalent components, defining candidate CRHs as tissue-specific elements equivalent to those found in CRHs but without 3D contacts. Among functional annotations with significant heritability, the heritability enrichment was higher for CRHs than for non-tissue-specific non-coding regions and candidate CRHs (Fig 5C), with strong enrichment signal for CRHs compared with candidate CRHs (Z-Score CRHs = 2.41, two-sided $P$-value = 0.01; Z-Score candidate CRHs = −1.84, two-sided $P$-value = 0.065). CRHs explained 11-fold more heritability than their respective candidate CRHs or up to 44-fold more than non-tissue–specific elements (Table S2). When compared with methods building hubs using only the chromatin contacts and using DNAse, CRHs built using the ABC approach performed better regarding schizophrenia heritability, showing enrichments of 3.08 (Fig 5C) against 0.84 (Fig S16A) and 2.98 (Fig S16B), respectively. All heritability enrichment results at the individual level and CRH level for the complete baseline model and his modified version are given for the ABC and control methods in Supplementary Tables (Tables S2 and S3). This result demonstrates a better concordance of the CRH including epigenetic features to explain schizophrenia heritability compared with only using chromatin interactions or combining chromatin interactions with chromatin accessibility.

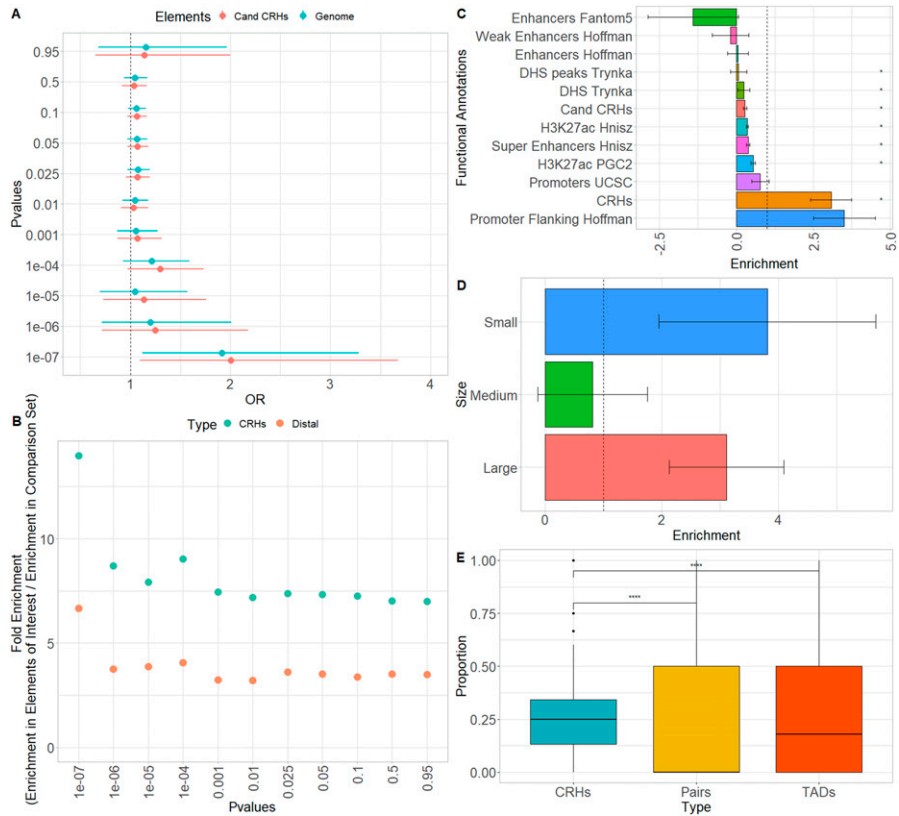

**Figure 5.  Cis-regulatory hubs (CRHs) are enriched in schizophrenia-associated SNPs, schizophrenia heritability, and capture links between noncoding SNPs and genes differentially expressed in schizophrenia.**
**(A)** SNP enrichment analysis measured through odds ratios (OR) and their 95% confidence interval for CRHs compared with candidate CRHs and the rest of the genome at different significance levels. The dotted line represents the null value. **(B)** Fold enrichment of distal elements and CRHs compared with their respective candidate sets for different significance levels. **(C)** Schizophrenia heritability enrichment measured with LDSC with error bars for CRHs, candidate CRHs, and non-tissue noncoding elements. The dotted line represents the null value. Errors bars represent the standard errors around the estimates of enrichment. **(D)** Schizophrenia heritability enrichment (measured with LDSC) with error bars for CRHs, considering the number of genes within CRHs. The dotted line represents the null value. **(E)** Boxplot of DEG proportions within CRHs, promoter distal elements pairs, and topologically associating domains. We considered only elements where we observed noncoding schizophrenia-associated SNPs. Differences were assessed using two-sided Wilcoxon rank-sum test. Data information: In (C) * represents $P$-value ≤ 0.05 after Bonferroni correction. In (E) **** represents $P$-values ≤ 2 × 10$^{-16}$.

Because we observed that schizophrenia-associated genes are highly expressed, enriched in small hubs, and connected to few distal elements, we defined strata of CRH number of promoters based on the proportion of total variance explained by CRHs (intraclass correlation) through a linear mixed model of the gene expression (Fig S17). Supporting our previous findings, we found that small CRHs (≤3 promoters) are more enriched in schizophrenia heritability than medium (>3 and ≤25 promoters) or large ones (>25 promoters) (Fig 5D). Overall, these results support that CRHs, especially small ones, are a relevant structure to explain the etiology of schizophrenia.

### CRHs predict the association between schizophrenia-associated noncoding SNPs and differentially expressed genes

Because we observed that CRHs are enriched in schizophrenia-associated SNPs and schizophrenia heritability, we wondered whether they represent a useful structure to link noncoding SNPs to genes differentially expressed in schizophrenia. To conduct this investigation, we leveraged information on 8,413 up- and down-regulated genes in all the available cell-types from a large set of schizophrenia patient brain tissues compared with controls (differentially expressed genes or DEGs) from SZBDMulti-Seq (Ruzicka et al, 2020 Preprint) and schizophrenia-associated SNPs from genome-wide association studies (The Schizophrenia Working Group of the Psychiatric Genomics Consortium et al, 2020 Preprint). First, CRHs were strongly enriched in DEGs compared with candidate genes, not included in CRHs (two-sided Fisher's exact test, odds ratio = 5.33, $P$-value ≤ 2 × 10$^{-16}$). CRHs encompassing at least one DEG exhibited a slightly larger proportion of

distal elements compared with CRHs without DEGs (median of 68% compared with 66%, two-sided Wilcoxon rank-sum test, $P$-value = 8.01 × 10$^{-6}$). These results suggest that CRHs may capture the links between SNPs in regulatory regions and DEGs. We tested the hypothesis that links between noncoding SNPs and DEGs are better captured by CRHs than by promoter-distal element pairs and TADs, respectively, the simplest form of CRHs and one of the most studied 3D feature in disease etiology (Bryois et al, 2018; Fudenberg & Pollard, 2019), by measuring the proportions of DEGs linked with noncoding SNPs. We first assigned schizophrenia-associated noncoding SNPs to each kind of structure (see the Materials and Methods section) and observed that 30% of CRHs exhibited at least one such assigned SNP in their regulatory regions, compared with 4%, and 95% for pairs and TADs, respectively. In the subset of elements where we observed assigned SNPs, CRHs exhibited a larger proportion of DEGs, exhibiting median proportion of 25% compared with 0% for regulatory regions directly connected with gene promoters and 18% for TADs, respectively (Fig 5E). Therefore, as intermediate structures compared with promoter-distal element pairs and TADs, CRHs better capture links between noncoding SNPs to gene expression variation possibly involved in schizophrenia.

## Discussion

Distal elements play a crucial role in complex diseases, such as schizophrenia. Recent studies have characterized relationships between promoters and distal elements interacting in hubs

(Madsen et al, 2020; Song et al, 2020; Espinola et al, 2021). However, their precise contributions to complex disease etiology remain unclear. In this study, we assessed the role of hubs linking promoters to distal elements in a complex disease. Thus, we defined CRHs in neurons as complex networks of gene promoters and distal elements in physical proximity (Fig 1A). CRHs aim to highlight direct and indirect contacts between promoters and distal elements which may not be targeted by other approaches. Our findings confirm the interest in integrating 3D contacts with tissue-specific regions to gain a deeper understanding of regulatory processes involved in complex diseases, where genetic disruptions may impact the transcription program of several genes (Figs 1C and S4). CRHs are enriched in gene promoters and distal elements associated with schizophrenia (Fig 5A) and explain a larger portion of heritability than candidate CRHs (Fig 5C) or other definitions to characterize CRHs (Fig S16A and B). Also, assessing the functional interest of CRHs in schizophrenia etiology, we found that CRHs are more efficient to capture the links between noncoding SNPs to genes differentially expressed in schizophrenia compared with TADs and promoter–distal element pairs. Thus, through CRHs, impacts of polymorphisms on gene expression variation can be better targeted. Therefore, our results establish that CRHs, by integrating interactions between distal elements and gene promoters, constitute a relevant 3D model to study complex diseases such as schizophrenia.

Previous studies suggest that hubs linking genes to enhancers are involved in the emergence of TADs (Espinola et al, 2021) or that highly interconnected enhancers constitute sub-TADs strongly enriched in CTCF (Madsen et al, 2020). Recent studies have either investigated the role of chromatin loops (Rajarajan et al, 2018) or the impact of ultra-rare variants in TAD borders in the emergence of schizophrenia (Halvorsen et al, 2020). CRHs constitute a more local functional organization than higher order chromatin features (A/B compartments, TADs) (Fig 2F and G) and are enriched in FIREs. In fact, CRHs are strongly enriched in active regions (Fig 3C), defining CRHs as functional hubs with high transcriptional activity. Moreover, CRHs are strongly enriched in schizophrenia-associated genes, which are characterized by higher expression levels (Fig 4B) and active regulatory regions (Fig 4D). These results are in line with those of Sey et al (2020), as they have shown that schizophrenia-associated genes exhibit higher differential expression in schizophrenia. Based on the above lines of evidence, we argue that focusing on CRHs should be prioritized over other levels of 3D organization in a context of complex phenotypes. Thus, through CRHs, impacts of polymorphisms on gene expression variation can be better targeted, aiming to highlight underlying regulatory processes.

Promoters and distal elements involved in CRHs exhibit different connectivity behaviors. Indeed, CRHs harbor more distal elements than genes (Fig 2C), suggesting that within a CRH, genes tend to have more connections compared with distal elements (Fig 2D) (Madsen et al, 2020; Espinola et al, 2021). Espinola et al (2021) have shown that hubs connecting promoters to distal elements encompass a single promoter, whereas Madsen et al (2020) exhibited that enhancers are mostly involved in one-to-one connections. These results suggest that genes have fewer specific relationships,

whereas enhancers, strongly connected to promoters, link genes with strong involvement in diseases (Madsen et al, 2020). However, in our data we found that CRH often harbor several genes connected by distal elements, supporting that CRHs can be either promoter hubs, enhancer hubs or multi hubs (Fig 2A) (Campigli et al, 2020). Limitations of CRHs defined from Hi-C data are their dependence on Hi-C resolution and the measure of contacts from multiple cells in bulk, which may lead to spurious merging of CRHs with contacts occurring in distinct cell sub-populations. Future studies using single-cell chromosome conformation will be needed to assess the relevance of CRHs at higher resolution (Nagano et al, 2013).

An important contribution of this study is to establish CRHs as a relevant model to study complex diseases such as schizophrenia. Indeed, we found strong enrichments in schizophrenia-associated SNPs, schizophrenia heritability within CRHs (Fig 5A and C), compared with candidate CRHs. Also consistent with this idea, we found that including DNAse hypersensitive sites and H3K27ac-enriched regions to the definition of CRH explains a larger portion of schizophrenia heritability than networks built only from chromatin contacts. Moreover, CRHs aim to highlight indirect connections between promoters and distal elements and our results show they offer an advantage over a pair of enhancer–promoter or larger domains to efficiently link noncoding SNPs to DEGs in schizophrenia. Collectively, these results point to the capability of CRHs to capture complex interplay between regulatory regions, which can help to fine map the functional regions involved in complex diseases, one of the most important challenges in polygenic diseases.

Moreover, schizophrenia-associated genes show fewer connections than non-associated ones and are enriched in monogamous relationships (Fig 4C and D). These results suggest that schizophrenia-associated genes are more strongly impacted than other active genes by disruptions of their distal elements because they are regulated by fewer connections to distal elements. Interestingly, we found that hubs encompassing a small number of genes highlight stronger schizophrenia heritability enrichments than medium or larger hubs (Fig 5D). We expect that small hubs or genes weakly connected to distal elements (monogamous, 1-1-N) will be more impacted by disruptions in their distal elements than large hubs or highly connected genes, supporting the model where weakly connected genes are more involved in disease etiology. From this study and others, the emerging model is that a gene with limited connections to distal elements will be more impacted by polymorphisms, whereas highly connected genes will have stronger environmental or genetic resilience to disruptions in their distal elements (Tsai et al, 2019).

Based on these results, we argue that CRHs capture direct and indirect connections between promoters and distal elements, explaining the underlying regulatory processes involved in complex phenotypes. Future studies will demonstrate whether CRHs as a functional 3D model improve detection power of causal genes or pathways to elucidate the underlying causal regulatory processes involved in complex diseases. Indeed, because a substantial portion of schizophrenia heritability remains to be explained, future work will be needed to assess the relevance of CRHs to help detect the rare variants which may be involved. CRHs can be integrated as functional annotation in association tests (He et al, 2017) or

proposed as new regions to aggregate variants in pathway-based approaches (Wu & Pan, 2018).

# Materials and Methods

### Hi-C data and pre-processing

Hi-C data for neurons from iPSCs at 10 Kb resolution were obtained from PsychENCODE Synapse platform (.hic format, intra-chromosomal). In the present study, we refer to these data as the neuron Hi-C dataset. Except for Score-FIRE calculation and ABC score, we applied KR-normalization with the Juicer toolbox (Durand et al, 2016) to obtain either a sparse or dense matrix.

### CRHs

CRHs were built based on the ABC model (Fulco et al, 2019) to capture active regulatory processes between distal elements and gene promoters. To validate analyses shown in the article, two other methods to build CRHs were also proposed (See Supplemental Data 1).

#### *ABC-Score*
The ABC model (Fulco et al, 2019) defines active enhancers based on a quantitative score of DNAse (ENCSR278FVO), H3K27ac (ENCSR331CCW), and normalized Hi-C contact number. This score is computed relative to a background activity over a 5-Mb window around a candidate element. Here candidate element refers to DNAse peaks on which enhancers are defined (Fulco et al, 2019). Then, we set the threshold to 0.012; beyond which a candidate element is considered as a distal element. This value was selected to ensure that the mean number of distal enhancers per promoter is between two and five in the neuron Hi-C dataset (Fulco et al, 2019).

As an extension of the ABC-Score, CRHs were defined as bipartite networks (igraph R package; Csardi & Nepusz, 2006) between promoters and distal elements. Because of the nature of the methodology of the ABC-Score, contacts between distal elements and promoters were restricted. In proposing CRHs based on the ABC-Score, active regulatory phenomena occurring in our tissue were captured.

CRHs are conceived to capture regulatory phenomena based on Hi-C. For the purpose of enrichment analysis for different external validation sources, SNPs or disease heritability, equivalent sets with elements having the same characteristics but in no 3D contacts with promoters were proposed. These elements were referred to as candidate CRHs. Thus, the same approach as Nasser et al (2021) was applied, where candidate distal elements are all DNAse peaks which do not overlap ABC distal elements. Also, candidate promoters were all promoters for known hg19 genes not included in CRHs.

### Summary statistics for schizophrenia

The original SCZ3 GWAS summary statistics (The Schizophrenia Working Group of the Psychiatric Genomics Consortium et al, 2020 *Preprint*) used in SNP and heritability enrichments were downloaded from the PGC site https://www.med.unc.edu/pgc/results-and-downloads. To assess independent SNPs in enrichment analyses, we used the clumped SNP file keeping the SNPs with the highest association signal with schizophrenia for a given genomic window.

### Schizophrenia-associated genes

To assess schizophrenia-associated genes, H-Magma (Sey et al, 2020) was used on iPSC-derived Hi-C neurons (Rajarajan et al, 2018) with schizophrenia SNP summary statistics (The Schizophrenia Working Group of the Psychiatric Genomics Consortium et al, 2020 *Preprint*) to link noncoding SNPs to their target gene. To determine significant schizophrenia-associated genes, all genes with a *P*-value lower or equal to 0.05 after Benjamini and Hochberg correction were selected.

### Partitioning heritability for schizophrenia

The LDSC regression (Finucane et al, 2015) was used to partition SNP heritability for schizophrenia integrating CRHs. For the main analysis, a modified version of the LDSC baseline model was used, only considering non-neuron-specific annotations corresponding to regulatory regions included within CRHs: promoters, H3K27ac histone marks, DNase I hypersensitive sites, ChromHMM/Segway predictions, super-enhancers, and FANTOM5 enhancers. By proceeding this way, we sought an unbiased comparison of SNP heritability in neuron CRHs compared with candidate CRHs or equivalent sets of genomic features widely used for this purpose. CRHs and candidate CRHs were extended by 500 bp upstream and downstream to consider the background activity and avoid inflating the enrichment signal, as suggested by Finucane et al (2015).

### Linking noncoding SNPs to DEGs in schizophrenia

To link noncoding SNPs to differentially expressed genes, we only considered the clumped SNPs because they represent the genetic variations the most associated with schizophrenia within genomic windows, without applying a *P*-value threshold. Then, the proportion of DEGs was calculated for the subset of CRHs, promoter distal–element pairs and TADs where we observed at least one SNP in one distal element.

### 3D features and other analyses

All analyses related to 3D features (A/B compartments, TADs, and FIREs), functional enrichments, peak calling, and other technical details are presented in Supplemental Data 1.

### Availability of data and materials

Datasets analyzed in this study are publicly available from: PsychENCODE Knowledge Portal (https://www.synapse.org, syn13363580, syn20833047) for Hi-C in iPSC, and postmortem brains, respectively. PGC3 (https://www.med.unc.edu/pgc/results-and-downloads) for summary statistics, SCREEN (https://screen.encodeproject.org) for Encode candidate regulatory elements in neural progenitor cell originated from H9, Roadmap

Epigenomics data Portal (https://egg2.wustl.edu/roadmap/data/byFileType/chromhmmSegmentations/ChmmModels/core_K27ac/jointModel/final) for 18-states model in E007, ENCODE data portal (ENCSR539JGB), and GEO (GSE142670) for reference epigenome and RNA-Seq in neurons. Various analyses shown in this article, as well as additional documentation and CRHs in neurons, are available at: https://github.com/lmangnier/Hi-C_analysis.

# Supplementary Information

# Acknowledgements

We would like to acknowledge Antoine Bodein, Julien Prunier, and Christophe Khun-Narom Tav of the Arnaud Droit Lab for important discussions and feedback during the preparation of this manuscript. We also thank Katie Pollard, Geoff Fudenberg, and Luis Chumpitaz-Diaz from the Pollard Lab for their insightful advice. Data were generated as part of the PsychENCODE Consortium, whose contributors and funding are listed in Supplemental Data 1. Funding: This work was partly funded by the Canadian Statistical Sciences Institute through a Collaborative Research Team grant and by a National Science and Engineering Research Council of Canada Discovery grant to A Bureau. Some data analyses were performed on computing resources from Compute Canada.

## Author Contributions

L Mangnier: conceptualization, data curation, software, formal analysis, validation, investigation, visualization, methodology, and writing—original draft, review, and editing.
C Joly-Beauparlant: validation, methodology, and writing—original draft, review, and editing.
A Droit: supervision and writing—original draft, review, and editing.
S Bilodeau: validation, methodology, and writing—original draft, review, and editing.
A Bureau: conceptualization, data curation, formal analysis, supervision, funding acquisition, investigation, methodology, project administration, and writing—original draft, review, and editing.

## Conflict of Interest Statement

The authors declare that they have no conflict of interest.

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
