## [Reviewer comments · Life Science Alliance]

Life Science Alliance

Cis-Regulatory Hubs: a new 3D Model of Complex Disease Genetics with an Application to Schizophrenia

Loic Mangnier, Charles Joly-Beauparlant, Arnaud Droit, Steve Bilodeau, and Alexandre Bureau

DOI: <https://doi.org/10.26508/lsa.202101156>

Corresponding author(s): Alexandre Bureau, Université Laval and Steve Bilodeau, Centre de Recherche du CHU de Québec - Université Laval

Review Timeline:

Submission Date:	2021-07-12
Editorial Decision:	2021-08-24
Revision Received:	2021-11-26
Editorial Decision:	2021-12-22
Revision Received:	2022-01-12
Accepted:	2022-01-14

Scientific Editor: Novella Guidi

Transaction Report:

August 24, 2021

Re: Life Science Alliance manuscript #LSA-2021-01156-T

Alexandre Bureau
Laval University

Dear Dr. Bureau,

Thank you for submitting your manuscript entitled "Cis-Regulatory Hubs: a Relevant 3D Model to Study the Genetics of Complex Diseases with an Application to Schizophrenia" to Life Science Alliance. The manuscript was assessed by expert reviewers, whose comments are appended to this letter. As you will note from the reviewers' comments below, the reviewers do raise some concerns. Specifically, Reviewer 1 main concerns is the lack of a clear introduction and solid establishment of model, requesting therefore to add diagrams to clearly demonstrate what ABC score is and the procedure of how CRHs are constructed using ABC scores. Another point is that authors do not provide solid validation of the significance of CRHs. This reviewer, thus, suggests showing the real advantages over a pair of enhancer-promoter or larger domains. Reviewer 2 main concern is about the lack of demonstration that the relatively small size of CHRs allows them to perform optimally their function compared to larger genome segments. For this point the reviewer suggest performing genome engineering by deleting or inserting potentially involved genomic elements and evaluating their biomolecular effects on the experimental system studied. Regarding this last experiment, in case you won't be able to perform it within the time frame for resubmission, please address the concern in the text by deeming down your statements. All the other concerns raised by the reviewers should be addressed as well. We, thus, encourage you to submit a revised version of the manuscript back to LSA that responds to all of the reviewers' points.

Thank you for this interesting contribution to Life Science Alliance. We are looking forward to receiving your revised manuscript.

Sincerely,

B. MANUSCRIPT ORGANIZATION AND FORMATTING:

Reviewer #1 (Comments to the Authors (Required)):

Mangnier et al. proposed a new concept, cis-regulatory hubs (CRHs), to study the gene and distal elements regulatory mechanism, with specific application to Schizophrenia. The authors introduced the CRHs through the description of genomic annotation features overlapping with the CRHs, such as A/B compartment, TADs, chromatin states. Further analysis with the Schizophrenia-associated genes, SNPs, and heritability demonstrated that schizophrenia-associated genes are within small hubs, with fewer connections to distal elements and high gene expression patterns.

The major concerns with this paper are

1. Lack of clear introduction and solid establishment of model, as well as the baseline methods or concepts that authors used for comparison.
2. Besides proposing the new model CRHs and describe the overlapping genomic feature and gene enrichment, the authors do not provide solid validation of the significance of CRHs. What kind of novel information or advantages that CRHs can bring.

Major issues:

1. Authors may consider adding a diagram figure to clearly demonstrate what ABC score is and the procedure of how CRHs are constructed using ABC scores. Even though the authors briefly explain the ABC score in the Methods section, but it is still very unclear to me how it is calculated. The other baseline methods that are used are also quite obscure, and authors can consider adding corresponding diagram figures for the Rao, DNase-based, and candidate CRH. Authors can also provide a genome browser view to visually show what does CRH look like and associate such genome browser view with the Schizophrenia genes/SNPs related findings.
2. No solid support or justification why bipartite graph is more suitable for this problem instead of using other more complicated graph models. Authors can do some literature reviews and benchmarked them to show these in the Results section can be trusted.
3. Since we already know the Schizophrenia genes, significantly relevant SNPs, ABC scores telling us the potential distal regulatory elements, what kind of new information or advantages can CRH brings? The authors discuss the potential advantages in the Discussion section. However, to convince the audience to realize the importance of CRH, authors need to show the real advantages over a pair of enhancer-promoter or larger domains. How do the indirect connections in the CRHs affect the gene regulation hence the Schizophrenia phenotypes? How does the complex epigenetic interplay? Or validation of the small hubs or weakly connected genes to distal elements is more impacted by disruptions hence more involved in disease etiology.

In other words, authors have focused too much on introducing what is the genomic feature and summarizing the statistics of CRHs, instead of showing how CRH can function to decipher the hub regulatory mechanism relating to Schizophrenia.

Minor issues:

1. Line 12 on page 5 has a weird period within the sentence.
"We defined CRHs using chromatin contacts provided by Hi-C data with and without additional epigenetic features defining classes of distal regulatory elements. to evaluate their role in schizophrenia etiology (See Supplementary Methods). "
2. Last line on page 5. How were the distal element and promoter expanded?
To build CRHs, the networks surrounding each distal element and promoter were expanded to include directly connected regions.
3. Last paragraph on page 6. Three "or" in a row makes it hard to understand what do authors really mean.
"1) the proportion of two or more distal elements sharing a promoter or two or more promoters sharing a distal element (i.e., 1-1-

N with $N > 0$) "

4. Authors may consider annotating Figure 2A right as a new panel and name it Figure2B.

5. For Figure 2C, I don't think authors can conclude that "CRHs are enriched in active distal elements ". Instead, Figure 2Aright has demonstrated that CRHs are enriched in Quies and inactive or repressor regions. Figure2C can only show that those CRHs overlapped with active state regions have higher connections (does more complex mean higher connections between CRH elements?)

6. Authors may check how the error bars are calculated for Figure 4C&D, some of which seem to have a very large variation. My understanding of these bar plots is that the bars represent the mean LDSR value for each category and the error bars are the standard deviation of the mean. Authors may check if they use the standard deviation of the mean or the sd of the population. (https://en.wikipedia.org/wiki/Standard_deviation#Standard_deviation_of_the_mean)

Reviewer #3 (Comments to the Authors (Required)):

COMMENTS

By exploiting as model human Schizophrenia, authors attempt to demonstrate that CHRs (Cis Regulatory Hubs), structures comprising genome segments important for transcription of genes, are involved in the pathogenesis of the disease. According to their report, their data demonstrate that genes and other genomic elements potentially causally involved in Schizophrenia' pathogenesis reciprocally contact themselves and activate previously silent pathological pathways. They further hypotesize that the relatively small size of CHRs allows them to perform optimally their function compared to larger genome segments. At this stage, this is just a hypothesis: it should explored through genome engineering by deleting or inserting potentially involved genomic elements and evaluating their biomolecular effects on the experimental system studied. Furthermore, post-mortem tissues should be analyzed to confirm data obtained mainly on a single iPSC line. Finally, the paper needs substantial editing.

Dear Dr. Guidi,

First of all, we would like to thank you and the reviewers for your time and constructive comments, which were used to improve and solidify the manuscript. We address all the major comments from the reviewers, giving us the opportunity to clarify certain aspects. Here is our point-by-point response to each comment.

Reviewer 1:

Major Comments:

1 “Authors may consider adding a diagram figure to clearly demonstrate what ABC score is and the procedure of how CRHs are constructed using ABC scores. Even though the authors briefly explain the ABC score in the Methods section, but it is still very unclear to me how it is calculated. The other baseline methods that are used are also quite obscure, and authors can consider adding corresponding diagram figures for the Rao, DNase-based, and candidate CRH. Authors can also provide a genome browser view to visually show what does CRH look like and associate such genome browser view with the Schizophrenia genes/SNPs related findings.”

Answer: We addressed these comments by providing new Figures (Fig1 and Figure in supplementary methods) illustrating both the ABC-Score and CRH building for each of our methods. In Fig1A, we described how CRHs were built from the ABC-Score, starting from the creation of functional pairs of promoters and active distal elements to CRHs. Also, Fig1B gives a concrete example of how functional pairs of elements are built based on H3K27ac, DNase and Hi-C contacts. Finally, in Fig1C, we provided a Genome Browser view to highlight what do CRHs look like, and their relation to schizophrenia-associated SNPs. For example, schizophrenia-associated SNPs can impact several genes by direct or indirect connections, possibly leading to gene dysregulation. This point is particularly relevant when we argue that through CRHs we can capture indirect connections between polymorphisms and disease-associated genes. We are convinced that adding these elements makes the method clearer for a large audience.

2 “No solid support or justification why bipartite graph is more suitable for this problem instead of using other more complicated graph models. Authors can do some literature reviews and benchmarked them to show these in the Results section can be trusted.”

Answer: We know that a large spectrum of methods have been proposed, specially to study networks of genes, interacting biologically (Barabasi & Oltvai. 2004). These methods have been extended to co-expression networks (Langfelder & Horvath 2008, van Dam et al. 2018). However, these methods require expression or differential expression for each gene, to compute a similarity measure between pairs of genes, which is not applicable in our context. Indeed, our proposed methods are built around physical contacts between a pair of biological elements (e.g., promoters and accessible regions) based on the Activity-by-contact Score. Thus, we proposed to build networks from these pairs. Also, even if recent papers consider promoter-promoter or enhancer-enhancer connections (Madsen et al. 2020, Song et al. 2020), the ABC-Score does not offer the capability to capture this kind of connections. In the light of these elements, we think that CRHs as bipartite networks are the more natural form of structure to highlight connections between promoters and distal elements. As an additional comment, we benchmarked the chosen method (e.g., ABC-Score) with two other methods (Rao-based and DNase-

based) with different level of biological annotations, capturing different complexities. We argue that applying these alternative methods can be viewed as sensitivity analysis, since we added additional biological layers to build CRHs and we compared results with respect to various approaches.

Barabási, AL., Oltvai, Z. Network biology: understanding the cell's functional organization. *Nat Rev Genet* **5**, 101–113 (2004). <https://doi.org/10.1038/nrg1272>

Langfelder, P., Horvath, S. WGCNA: an R package for weighted correlation network analysis. *BMC Bioinformatics* **9**, 559 (2008). <https://doi.org/10.1186/1471-2105-9-559>

Sipko van Dam, Urmo Vösa, Adriaan van der Graaf, Lude Franke, João Pedro de Magalhães, Gene co-expression analysis for functional classification and gene–disease predictions, *Briefings in Bioinformatics*, Volume 19, Issue 4, July 2018, Pages 575–592, <https://doi.org/10.1093/bib/bbw139>

3 “Since we already know the Schizophrenia genes, significantly relevant SNPs, ABC scores telling us the potential distal regulatory elements, what kind of new information or advantages can CRH brings? The authors discuss the potential advantages in the Discussion section. However, to convince the audience to realize the importance of CRH, authors need to show the real advantages over a pair of enhancer-promoter or larger domains. How do the indirect connections in the CRHs affect the gene regulation hence the Schizophrenia phenotypes? How does the complex epigenetic interplay? Or validation of the small hubs or weakly connected genes to distal elements is more impacted by disruptions hence more involved in disease etiology.”

Answer: The reviewer is raising important questions which are highly relevant. To provide more evidence of the importance of CRHs, we performed additional analyses linking non-coding SNPs to differentially expressed genes using CRHs, promoter-distal element pairs and TADs, respectively. Briefly, our analyses demonstrated that CRHs provide a more efficient structure to target impacts of polymorphisms on differentially expressed genes in schizophrenia than simple promoter-distal element pairs and TADs (Fig 5E). Moreover, regarding the capability of CRHs to encompass indirect connections between polymorphisms and central genes in diseases (e.g., epigenetic interplay), we found that CRHs exhibiting differentially expressed genes have higher proportions of distal elements than CRHs without. This result supports the idea that CRHs is a suitable model to study direct but more likely indirect epigenetic mechanisms that may be leading to a disease. To report all these new results, we added a new section in the paper called “CRHs predict the association between schizophrenia-associated non-coding SNPs and differentially expressed genes” on page 20.

Minor Comments:

1. “Line 12 on page 5 has a weird period within the sentence.

‘We defined CRHs using chromatin contacts provided by Hi-C data with and without additional epigenetic features defining classes of distal regulatory elements. to evaluate their role in schizophrenia etiology (See Supplementary Methods).’

Answer: The typo was corrected directly in the paper

2. “Last line on page 5. How were the distal element and promoter expanded?

To build CRHs, the networks surrounding each distal element and promoter were expanded to include directly connected regions.”

Answer: We clarified the sentence. It now states "... we identified 62,658 functional pairs of distal elements and promoters where the ABC score exceeded a threshold of 0.012. The value of the threshold was chosen so that we had on average, 4.51 distal elements per genes, as recommended by Fulco et al. 2019. CRH were built from these connections between promoters and distal elements". The new figure 1 should help to clarify this aspect.

3. "Last paragraph on page 6. Three "or" in a row makes it hard to understand what do authors really mean.

'1) the proportion of two or more distal elements sharing a promoter or two or more promoters sharing a distal element (i.e., 1-1-N with $N > 0$) "'

Answer: The sentence was replaced by "the proportion of instances where one distal element connects one promoter with at-least one other distal element or the reverse: one promoter connects one distal element with at least one other promoter (i.e., 1-1-N with $N > 0$)"

4. Authors may consider annotating Figure 2A right as a new panel and name it Figure2B.

Answer: We divided Figure 2A into two distinct figures (now Figures 3A and 3B) to make the reading easier for the reader.

5. "For Figure 2C, I don't think authors can conclude that "CRHs are enriched in active distal elements ". Instead, Figure 2Aright has demonstrated that CRHs are enriched in Quies and inactive or repressor regions. Figure2C can only show that those CRHs overlapped with active state regions have higher connections (does more complex mean higher connections between CRH elements?)"

Answer: Throughout the paper, enrichments are computed by comparing CRHs with candidate CRHs. Briefly, candidate CRHs are tissue-specific elements not integrating 3D contacts. We agree with the fact that a large proportion of CRHs overlap Quies and inactive or repressor regions. However, comparing CRHs to their comparison set (i.e., candidate elements), we found that they exhibit larger proportions of active elements compared to candidate CRHs.

6. Authors may check how the error bars are calculated for Figure 4C&D, some of which seem to have a very large variation. My understanding of these bar plots is that the bars represent the mean LDSR value for each category and the error bars are the standard deviation of the mean. Authors may check if they use the standard deviation of the mean or the sd of the population.

https://en.wikipedia.org/wiki/Standard_deviation#Standard_deviation_of_the_mean

Answer: Indeed, we did not specify clearly how did we obtain error bars when heritability enrichments are considered, possibly creating confusion. Following the methodology presented in Finucane et al. 2015, error bars were obtained with standard errors around the estimates of enrichment for a given annotation. We clarified this point in the legend of Figure 5.

Reviewer 3:

"By exploiting as model human Schizophrenia, authors attempt to demonstrate that CHRs (Cis Regulatory Hubs), structures comprising genome segments important for transcription of genes, are involved in the pathogenesis of the disease. According to their report, their data demonstrate that genes and other

genomic elements potentially causally involved in Schizophrenia' pathogenesis reciprocally contact themselves and activate previously silent pathological pathways. They further hypothesize that the relatively small size of CHR allows them to perform optimally their function compared to larger genome segments. At this stage, this is just a hypothesis: it should explore through genome engineering by deleting or inserting potentially involved genomic elements and evaluating their biomolecular effects on the experimental system studied.”

Answer: Even if genome engineering would provide experimental justification of CHR, our main objective here is to provide a valid computational model regarding biological, epigenetic, and genetic associations. While we are not including genome engineering experiments, we are adding additional layers of validation of the model. Indeed, we functionally validate CHR through the link between non-coding SNPs and differentially expressed genes in schizophrenia, compared to direct pairs of promoter-distal elements and TADs (See answer #3 to reviewer 1). These results can be found in the new section on page 20 “CHR predict the association between schizophrenia-associated non-coding SNPs and differentially expressed genes”.

“Furthermore, post-mortem tissues should be analyzed to confirm data obtained mainly on a single iPSC line.”

Answer: We thank the reviewer for this relevant suggestion. We built CHR and validated them regarding complexity and relationship with 3D features in dopaminergic neurons and a general neuron sample from post-mortem brain tissues from three individuals. First, our results point to a direct or indirect sharing of most pairs of promoters and distal elements between CHR built from iPSC and from post-mortem brain (even though the overlap was not perfect). Then, regarding CHR structures and complexity, we found that in post-mortem brain, CHR are mostly composed by distal elements where promoters tend to connect more than one distal element, confirming our results in iPSC-derived neurons. Finally, we assessed the link between CHR and 3D features and found that in post-mortem brain CHR represent more local 3D structures than A/B compartments or TADs. Results on post-mortem brain are shown in Figures S1 to S8.

“Finally, the paper needs substantial editing.”

Answer: The paper was reviewed thoroughly and edited to improve readability.

In addition to the new analyses and edits made in response to reviewer comments, certain analyses were redone to validate the results. This led to the following modifications:

- First, to standardize statistical tests, we replaced right-tailed with two-sided tests to consider alternative hypotheses contrary to expectation (e.g. depletion instead of enrichment); p-values were recomputed accordingly. It is worth noting that some effect measures are slightly changed to the second decimal point.
- Then, in the chromatin state analysis, we adjusted definitions of overlapping patterns to obtain a more intuitive interpretation. Indeed, in the original submission, only distal elements were considered while in this revision we integrated both distal elements and promoters. We changed “We found that the majority of CHR elements (72%) overlapped at least one Inactive or Repressor region against 53% and 39% of CHR elements for Active or Weakly Active regions, respectively (Figure 2A left)” with “At the broad category level, we found that most elements

(promoters and distal elements) included in CRHs (58%) overlapped Weakly Active regions against 49% for Inactive or Repressor and 53% for Active regions, respectively (Fig 3A).”, adjusting the subsequent figure (Fig 3A).

- Finally, regarding the logistic regression performed in the section “Multivariate analysis of CRH features with respect to schizophrenia-associated genes” on page 16 and the subsequent Fig4D, we found duplicated entries in the dataset. Removing duplicates led to slightly different results (confidence intervals and estimates) without changing substantially the interpretation of the variable effects.
- The axis labels of Figure 4C now correctly show percentages, and the layout of Figure 4 was improved.

We also complied with the format requirements of LSA, for instance adding a summary blurb and figure titles and shortening the title and abstract.

In conclusion, we believe this version of the manuscript is much improved, validating the biological significance of CRHs in schizophrenia. We hope you will now deem it acceptable for publication in LSA.

Sincerely,

Alexandre Bureau, PhD

Department of social and preventive medicine, Laval University

CERVO Brain Research Centre

2601, chemin de la Canardière

Québec, Québec

Canada

G1J 2G3

E-Mail alexandre.bureau@fmed.ulaval.ca

December 22, 2021

RE: Life Science Alliance Manuscript #LSA-2021-01156-TR

Dr. Alexandre Bureau
Université Laval
Médecine sociale et préventive
1050 rue de la Médecine
Québec G1V 0A6
Canada

Dear Dr. Bureau,

Thank you for submitting your revised manuscript entitled "Cis-Regulatory Hubs: a new 3D Model of Complex Disease Genetics with an Application to Schizophrenia". We would be happy to publish your paper in Life Science Alliance pending final revisions necessary to meet our formatting guidelines.

- as pointed by Reviewer 1, please fix the blurry figures, especially figure 1C and add similar examples in the supplementary to gain more audience interest in the field
- please add the Twitter handle of your host institute/organization as well as your own or/and one of the authors in our system
- please use the [10 author names, et al.] format in your references (i.e. limit the author names to the first 10)
- please add callouts for Figures S1A, B; S4A-C; S5A-C; S9A, B; S10A, B; S11A, B; S12A, B; S13A, B to your main manuscript text

A. FINAL FILES:

B. MANUSCRIPT ORGANIZATION AND FORMATTING:

Sincerely,

Reviewer #1 (Comments to the Authors (Required)):

The authors have added sufficient figures to demonstrate the model and included more analysis to gain biological insights. The authors have addressed all my questions.
One minor concern is that the figures are very blurred on my side. Figure 1C is very informative and enlightening, but it is impossible to see the genes and risk variants information in the current version. Also, authors may consider adding more similar examples in the supplementary to gain more audience interest in the field.

Reviewer #3 (Comments to the Authors (Required)):

Authors did convincingly reply to my original comments and requests: accordingly, in my opinion this paper may now be accepted for publication.

January 14, 2022

RE: Life Science Alliance Manuscript #LSA-2021-01156-TRR

Dr. Alexandre Bureau
Université Laval
Médecine sociale et préventive
1050 rue de la Médecine
Québec G1V 0A6
Canada

Dear Dr. Bureau,

Thank you for submitting your Research Article entitled "Cis-Regulatory Hubs: a new 3D Model of Complex Disease Genetics with an Application to Schizophrenia". It is a pleasure to let you know that your manuscript is now accepted for publication in Life Science Alliance. Congratulations on this interesting work.

DISTRIBUTION OF MATERIALS:

Again, congratulations on a very nice paper. I hope you found the review process to be constructive and are pleased with how the manuscript was handled editorially. We look forward to future exciting submissions from your lab.

Sincerely,
